# Virtual reality vs. Tablet video for venipuncture education in children: A randomized clinical trial

**Jiyoun Lee**[ORCID]1‡, **Jung-Hee Ryu**1,2‡, **Soo Hyun Seo**3, **Sunghee Han**1,2‡*, **Jin-Woo Park**[ORCID]1,2‡*

1 Department of Anesthesiology and Pain Medicine, Seoul National University Bundang Hospital, Seongnam, Korea, 2 Department of Anesthesiology and Pain Medicine, College of Medicine, Seoul National University, Seoul, Korea, 3 Department of Laboratory Medicine, Seoul National University Bundang Hospital, Seongnam, Korea

‡ JL and JHR are contributed equally to this project as co-first authors. SH and JWP are contributed equally to this project as co-corresponding authors.
* jinul8282@gmail.com (JWP); anesthesia@snubh.org (SH)

## Abstract

Pediatric patients usually experience high levels of pain and distress due to venipuncture. This randomised study aimed to evaluate the effects of virtual reality-based preprocedural education in comparison with video-based education in terms of pain and distress experienced by children scheduled to undergo venipuncture. Ninety children aged 4–8 years who were scheduled to undergo venipuncture surgery were randomly assigned to either a video or virtual reality group. Children in the video group received preprocedural education on venipuncture via a video displayed on a tablet and those in the virtual reality group received the same education via a head-mounted virtual reality display unit. The educational content for the two groups was identical. An independent assessor blinded to the group assignment observed the children's behavior and determined their Children's Hospital of Eastern Ontario Pain Scale scores, parental satisfaction score, procedure-related outcomes, venipuncture time, number of repeated procedures and difficulty score for the procedure. The virtual reality group experienced less pain and distress, as indicated by their Children's Hospital of Eastern Ontario Pain Scale scores compared with the video group (5.0 [5.0–8.0] vs. 7.0 [5.0–9.0], $P = 0.027$). There were no significant intergroup differences in parental satisfaction scores or procedure-related outcomes. For pediatric patients scheduled to undergo venipuncture, preprocedural education via a head-mounted display for immersive virtual reality was more effective compared with video-based education via a tablet in terms of reducing pain and distress.

## Introduction

Venipuncture is one of the most distressing and painful medical procedures for pediatric patients [1]. Unpleasant and painful experiences may contribute to high anxiety levels before the procedure and result in uncooperative behavior during the procedure [2, 3]. Additionally,

**Data Availability Statement:** According to our institution's IRB (Institutional Review Board) regulations, we cannot upload the data from our research to a public server. The IRB will act as the contact point for determining appropriate requests

for data sharing. The contact information is
snubhirb@gmail.com, with an IRB approval
number: B-2211-791-301.

**Funding:** This work was supported by the National
Research Foundation of Korea (NRF) grant funded
by the Korea government (MSIT). (No.
2022R1F1A1072210). The funders had no role in
study design, data collection and analysis, decision
to publish, or preparation of the manuscript.

**Competing interests:** Sunghee Han and Jin-Woo
Park are the co-inventors of the patent, "Medical
experience in hospitals provided with VR or AR
system", application of which is pending.

pain from medical procedures may result in long-lasting negative effects among pediatric patients, including the development of fear in adulthood, avoidance of medical care, missed medical appointments, and inadequate health care follow-up [3]. Therefore, it is crucial to ensure effective management of pain and distress during venipuncture procedures for the physical and psychological well-being of pediatric patients [4].

Non-pharmacological or behavioral interventions are necessary to reduce pain and distress during venipuncture because pharmacological interventions are typically administered through the intravenous line after venipuncture. Among non-pharmacological behavioral strategies, pre-venipuncture education via various interventions, such as preparation programs and procedural information, has been reported to relieve procedure-related pain and distress in children [5–7]. These behavioral approaches to pediatric medical pain are based on the gate control theory, which explains the transmission and modulation of pain signals [3, 8]. According to the gate control theory, pain signals can be inhibited by the closing of "gates" in the spinal cord and familiarity with preprocedural education may reduce anxiety and distress, which in turn decreases pain during painful medical procedures in children [3].

Advancements in information technology (IT) have enabled the application of immersive virtual reality (VR) in pediatric patients, and clinical studies related to VR-based education for pediatric patients are rapidly increasing so as to alleviate anxiety and distress [9–15]. Studies on VR-based education have outperformed standard care involving the communication of simple verbal information in terms of mitigating anxiety and distress among pediatric patients before anesthesia or non-invasive procedures [9, 10, 12–14]. Investigations on the effect of VR as a preprocedural educational tool for painful procedures, such as venipuncture, showed that VR is useful in reducing pain and anxiety compared with standard care involving the communication of simple verbal information [11, 15]. Furthermore, the audio–visual parts included in VR-based education may have caused the observed effect irrespective of VR.

Therefore, we hypothesized that preprocedural education with more immersive VR could more effectively decrease the pain and discomfort experienced by children during venipuncture compared with tablet videos conveying the same content but without a VR component. To this end, in this prospective randomized controlled trial, we evaluated the pain and distress of pediatric patients during venipuncture and procedure-related outcomes after they received VR- or video-based preprocedural education with identical content. To our knowledge, this is the first such study aimed at elucidating the authentic effect of immersive VR-based education on venipuncture-related pain and distress through a comparison with education via a tablet video.

## Methods

### Study design

This randomized clinical trial was approved by the institutional review board (IRB) of Seoul National University Bundang Hospital (SNUBH; IRB number: B-2211-791-301; approval date: October 24, 2022). The protocol was registered in the University Hospital Medical Information Network Clinical Trials Registry (registration number: UMIN 000049307; registration date: October 25, 2022). Written informed consent was obtained from the parents of children younger than 7 years of age and from both a parent and the child for children aged 7 years or higher. This prospective study was performed from October 31, 2022, to April 20, 2023, at SNUBH.

### Patients

This study included children aged 4–8 years who were scheduled to undergo venipuncture at the phlebotomy unit of SNUBH. Children with congenital problems, hearing or vision impairments, intellectual developmental difficulties, cognitive deficiency, seizure history,

psychoactive medicine prescriptions, or a history of venipuncture in the previous year were excluded from the study. Of the 127 children assessed for eligibility, 37 were excluded owing to their refusal to participate. The remaining 90 children participated in the study; none dropped out (Fig 1).

## Randomization

Using a computer-generated randomization code (Random Allocation Software, version 1.0; Isfahan University of Medical Sciences), the enrolled participants were randomly assigned in a 1:1 ratio to either a VR or a video group. An independent researcher performed this randomization 10 min before venipuncture. The researcher also asked the patients to predict procedural pain by assigning a score from 0 to 10 by using a visual analogue scale (score 0: no pain; score 10: the worst pain). Another independent researcher received a sealed envelope with the randomization number and performed the intervention in an independent room separated from the phlebotomy unit.

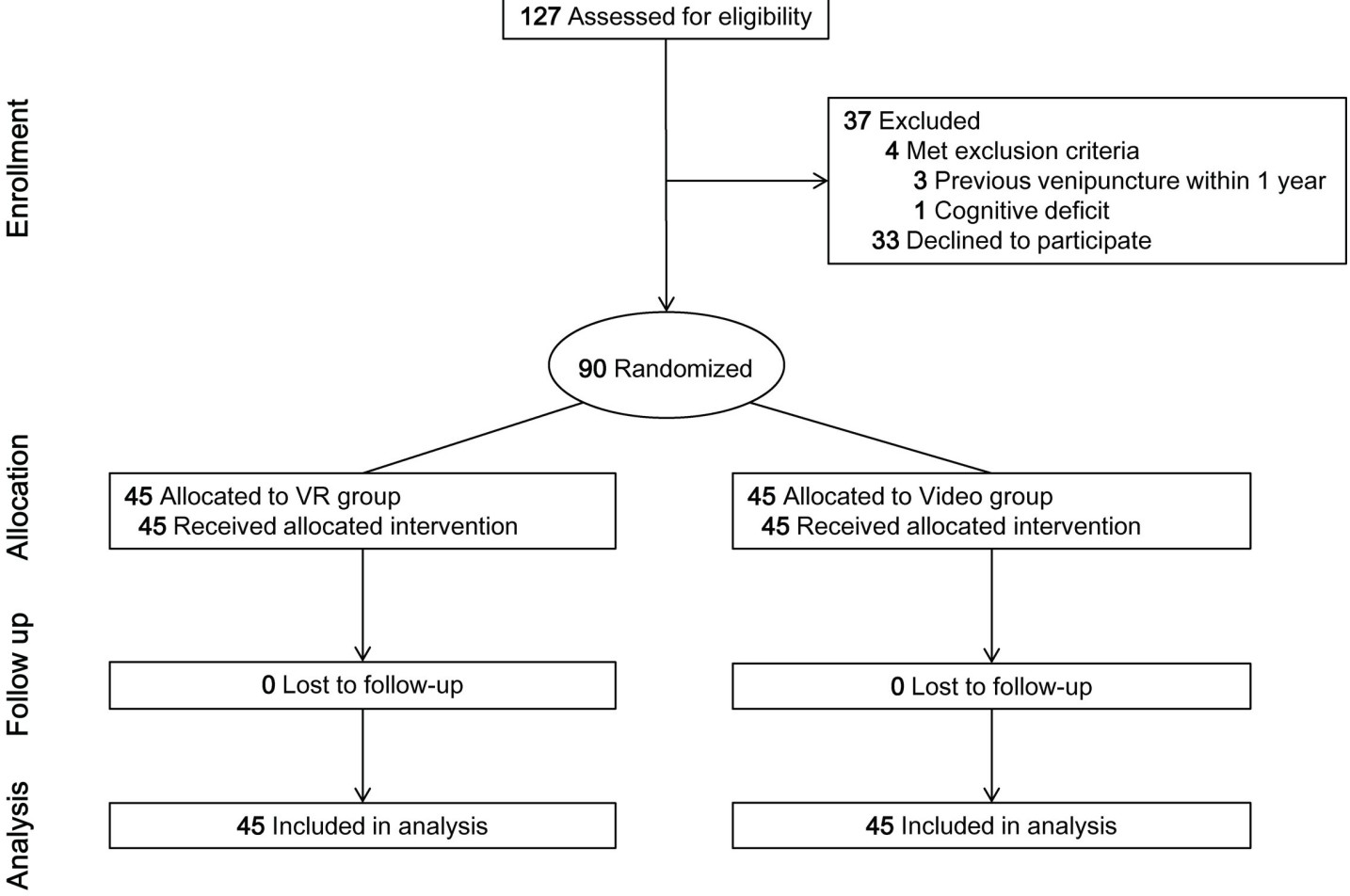

**Fig 1. CONSORT flow diagram.** VR, virtual reality.

## Intervention

The VR group received VR-based preprocedural education for 4 min, as described by Ryu and co-workers. In brief, cartoon characters from "Hello Carbot" (a famous Korean animation movie; ChoiRock Contents Factory, Seoul, South Korea) welcomed the children at the phlebotomy unit in a 360° three-dimensional virtual universe. After the patient chose one of the characters based on his/her preference, the character kindly explained the purpose and process of venipuncture to the child. The child also experienced venipuncture at a virtual phlebotomy desk and learned to position himself or herself appropriately during the procedure. The cartoons enthusiastically encouraged the child to cooperate properly (Fig 2A). We secured the permission to use the cartoon characters through a licensing agreement with ChoiRock Contents Factory. The virtual education was provided using MetaQuest 2 (Meta, Menlo Park, CA, USA; Fig 2B), the graphics quality in which was superior to that in the previously used version. The content was produced in partnership with a VR software development company (Formal-Works, Inc., Seoul, South Korea). The video group received video-based education for 4 min via a tablet (iPad, Apple Inc., Cupertino, CA, USA; Fig 2C). The content was identical to that used for the VR group, i.e., the content used in the VR group was transformed into a two-dimensional video.

## Study outcomes

Immediately after the VR or video session, the patients were moved to the phlebotomy unit for venipuncture. The interval from the end of education to the positioning at the phlebotomy desk did not exceed 5 min. An independent assessor blinded to the group assignment observed the children's behavior and determined the Children's Hospital of Eastern Ontario Pain Scale (CHEOPS) scores [16]. The CHEOPS score, the primary outcome of this study, was calculated from the scores for each of the six categories: crying, facial expression, verbal response, torso, hands, and legs (score range: 4–13; S1 Table). The scores were proportional to the children's pain and distress levels [16].

The total time for venipuncture (from positioning at the phlebotomy desk to needle insertion for successful blood sampling) and the incidence of repeated venipunctures performed owing to poor cooperation were recorded. After the procedure, the assessor asked the participants' parents to indicate their satisfaction with the venipuncture process by using a numerical rating scale (NRS; score 0 = extremely dissatisfied; score 10 = extremely satisfied). Immediately after the patients and parents exited the phlebotomy unit, the phlebotomy technicians were asked to evaluate their perception of the difficulty experienced in performing the procedure by using an NRS (score 0 = extremely easy; score 10 = extremely difficult).

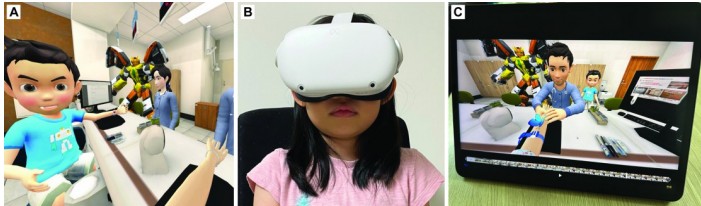

**Fig 2. Virtual reality education system.** (A) A cartoon character enthusiastically encourages the participant not to be anxious during a virtual experience of venipuncture procedure. (B) A head-mounted virtual reality display, Meta Quest 2. (C) A tablet PC, iPad, for video education. Reprinted from ["Hello Carbot"] under a CC BY license, with permission from [ChoiRock Contents Factory], original copyright [2017].

## Statistical analysis

Continuous variables are indicated as median (interquartile range) or mean (standard deviation) values according to the normality of the data. Categorical variables are presented as numbers (%). The Mann–Whitney U test was used to compare continuous study outcomes between the VR and video groups. Categorical outcomes were compared between the study groups by using Fisher's exact test or the chi-square test, as appropriate. Multiple linear regression analysis was performed to determine independent factors associated with the CHEOPS pain score. All statistical analyses were performed using SPSS software (version 21.0; SPSS Inc., Chicago, IL, USA). Statistical significance was defined as a two-sided $P$-value of $<0.05$.

## Sample size

In a pilot study of 40 pediatric patients (20 pairs) undergoing venipuncture, the CHEOPS scores (mean [standard deviation]) for the video and VR groups were 7.7 (2.0) and 6.5 (1.7), respectively. Power analysis was performed using G*Power 3.1.2 (Heinrich-Heine University, Düsseldorf, Germany). Based on the pilot data, a sample size of 45 patients per group was calculated to be necessary, with a power of 0.8, significance level of 0.05, and an assumed dropout rate of 10%.

## Results and discussion

Patient characteristics were comparable between the VR and video groups. The venipuncture-related pain expected by the participants before the procedure was similar between the two groups (Table 1).

Children's pain and distress assessed using CHEOPS were significantly lower in the VR group (median [interquartile range, IQR], 5.0 [5.0–8.0]) than in the video group (median [IQR], 7.0 [5.0–9.0]) ($P = 0.001$; Table 2). When the pain score was classified into three grades (mild, CHEOPS score of 4–6; moderate, CHEOPS score of 7–10; and severe CHEOPS score of 11–13), the proportion of children with mild procedural pain was significantly higher in the VR group than in the video group ($P = 0.02$; S1 Fig). Parental satisfaction with the venipuncture procedure and procedure-related outcomes, including procedure time, incidence of repeated venipuncture, and procedural difficulty score evaluated by the phlebotomy technicians, were not significantly different between the two groups (Table 2).

Multiple linear regression analysis revealed that the expected pain score before the procedure (measured using the visual analogue scale [VAS]) and each group (VR or video) were independent predictors of the CHEOPS score during venipuncture (Table 3).

This study revealed that immersive VR is more effective than non-immersive videos for preprocedural education regarding procedural pain and distress in pediatric patients undergoing venipuncture. To the best of our knowledge, this is the first study to demonstrate that

**Table 1. Patients' characteristics.**

|  | Tablet ($n = 45$) | VR ($n = 45$) | $P$ value |
|---|---|---|---|
| Age, median (IQR), years | 7.0 (5.0–8.0) | 7.0 (6.0–8.0) | 0.250 |
| Male, No. (%) | 22 (48.9%) | 26 (57.8%) | 0.398 |
| Weight, median (IQR), kg | 23.0 (19.7–28.7) | 25.0 (23.0–29.0) | 0.059 |
| Height, mean (SD), cm | 120.9 (9.9) | 124.4 (8.4) | 0.174 |
| Preprocedural pain expectation, median (IQR), VAS | 4.0 (0.0–6.0) | 4.0 (2.0–6.0) | 0.910 |

IQR, interquartile range; SD, standard deviation; VR, virtual reality; VAS, visual analogue scale.

**Table 2. CHEOPS, parental satisfaction, and procedural data.**

|  | Video (*n* = 45) | VR (*n* = 45) | *P* value |
|---|---|---|---|
| CHEOPS, median (IQR) | 7.0 (5.0–9.0) | 5.0 (5.0–8.0) | 0.027 |
| Parental satisfaction score, median (IQR), NRS | 10.0 (9.0–10.0) | 10.0 (9.0–10.0) | 0.721 |
| Venipuncture time, median (IQR), s | 41.1 (29.6–55.0) | 40.2 (28.0–48.8) | 0.519 |
| Repeated Procedures, No. (%) | 2 (4.4%) | 4 (8.9%) | 0.677 |
| Difficulty score for the procedure, median (IQR), NRS | 1.0 (0.0–5.0) | 0.0 (0.0–2.0) | 0.369 |

CHEOPS, Children's Hospital of Eastern Ontario Pain Scale; IQR, interquartile range; NRS, numeric rating scale; VR, virtual reality.

immersive VR *per se* is an effective tool for preprocedural education in pediatric patients undergoing painful medical procedures. Our finding of immersive VR being more effective than video in preprocedural education is in line with previous pedagogical research reports. Immersive VR is a more effective tool than nonimmersive videos in pre-training education because immersive VR increases enjoyment, intrinsic motivation, and knowledge transfer [17–20]. Our results, which are in line with those of previous studies, suggest that the pedagogical strength of immersive VR can be adopted in preprocedural education.

Most previous studies that presented the positive effects of VR in pediatric medical care failed to show the effects of VR *per se*. This is because these studies investigated the effectiveness of VR by comparing the content presented via VR and those presented as part of standard care. The VR content used in these studies consisted of a combination of audio and visual components. However, standard care involved education via simple verbal communication. It is well known that multiple sensory modalities (e.g., audio and visual stimulation together) enhance learning more effectively compared with a single modality (e.g., audio or visual stimulation alone) and that multimedia education outperforms verbal education in terms of learners' behavior change or knowledge achievement, which is called the modality effect theory [21, 22]. In this regard, the most recent research regarding the effect of immersive VR in pediatric patients may have proven the effect of multimedia or the modality effect rather than the effect of immersive VR technology itself. In this study, we excluded any confounding bias related to educational modalities or quality of educational content by using identical audio–visual information in the two study groups.

Several studies have compared the effects of immersive VR and video in pediatric patients. However, none of these studies have revealed the effect of immersive VR on the preprocedural education of pediatric patients. Most of these studies targeted patient distractions but not education [23–27]. In addition, some studies adopted different content between the VR and video groups [26, 27]. To the best of our knowledge, only one study has compared the effects of

**Table 3. Multiple linear regression analysis: Predictors of CHEOPS score.**

|  | B | 95% CI | Beta | *P* value |
|---|---|---|---|---|
| Age | -0.090 | -0.544–0.363 | -0.050 | 0.693 |
| Height | 0.021 | -0.060–0.102 | 0.089 | 0.604 |
| Weight | -0.062 | -0.167–0.042 | -0.175 | 0.238 |
| Gender | -0.569 | -1.394–0.255 | -0.130 | 0.173 |
| Pain expectation (VAS) | 0.345 | 0.223–0.468 | 0.517 | < 0.001 |
| Education group | -0.890 | -1.683 – -0.096 | -0.203 | 0.028 |

CHEOPS, Children's Hospital of Eastern Ontario Pain Scale; VAS, visual analogue scale.

immersive VR and video on patient education by evaluating the effect of VR *per se* in the chest radiography setting [14]. In contrast to the present study, this previous study involved non-invasive and painless procedures [14]. The present study demonstrated the effectiveness of immersive VR in pediatric patients undergoing painful and distressing medical procedures.

The multiple linear regression analysis in our study also suggested that the expected pain scores before the procedure and the group assigned were independent predictors of the CHE-OPS score during venipuncture. Our findings are consistent with those of previous studies that reported positive associations between anticipatory anxiety and procedural pain in children [28]. They found that anticipatory anxiety was strongly associated with pain intensity through various stimuli, including thermal, pressure, and cold pain tasks, in healthy children and adolescents [28].

The total time for venipuncture and the number of venipunctures performed due to poor cooperation were recorded. After the procedure, the phlebotomy technicians were also asked about the subjective difficulty experienced when performing venipuncture. All procedure-related outcomes were comparable between the two groups, which contradicts the results of a previous investigation involving chest radiography, a non-invasive and painless procedure [14]. During chest radiography, the procedure time and degree of difficulty for the radiologist were lower in the VR group than in the tablet group [14]. This may be explained by the difference between painful and painless procedures.

The parents of the participants were asked about their satisfaction with the venipuncture process; there was no significant difference in this regard between the two groups. The mean score of parental satisfaction was 10 in both groups, indicating that the parents were highly satisfied with the preprocedural education using VR or tablets. These results are contrary to those of a previous study that showed a significant difference in parental satisfaction between the VR group and the control group which received simple verbal education [15]. Active multimedia education to reduce procedural pain in pediatric patients might increase the satisfaction of the parents, in both groups.

Our study has some limitations. First, a considerable number of children did not want to participate in this study; most of them were reluctant to wear a head-mounted display for a virtual experience. Since VR devices are not yet widespread, children experiencing them for the first time may be reluctant to wear them. Therefore, the study only included children who had a positive attitude toward the VR experience and were expected to show positive outcomes, which may have resulted in selection bias. However, in our study, a blinded researcher performed the observational measurements; selection would not significantly bias our conclusions for children who received VR or video education. Second, the VR experience has been reported to cause complications, such as motion sickness and eye strain [29, 30]. Therefore, researchers should try to prevent these side effects and manage them appropriately. In this study, no complications due to the virtual education, which might be attributed to the relatively short play time and minimal virtual movement during the experience. Compared with adults, children are also known to experience fewer side effects from the VR experience [31].

## Conclusions

In conclusion, this randomized controlled trial showed that immersive VR using a head-mounted display is more effective than a non-immersive video for preprocedural education regarding procedural pain and distress in pediatric patients receiving venipuncture.

## Supporting information

**S1 Fig. Distribution of pain grades based on CHEOPS score.** CHEOPS, Children's Hospital of Eastern Ontario Pain Scale.
(TIF)

**S1 Table. Children's Hospital of Eastern Ontario Pain Scale (CHEOPS).**
(DOCX)

**S1 Checklist. CONSORT 2010 checklist of information to include when reporting a randomised trial\*.**
(DOC)

**S1 Protocol. Study protocol.**
(DOCX)

## Author Contributions

**Conceptualization:** Jung-Hee Ryu, Sunghee Han.

**Data curation:** Jiyoun Lee, Sunghee Han.

**Formal analysis:** Jiyoun Lee, Jung-Hee Ryu, Soo Hyun Seo, Sunghee Han, Jin-Woo Park.

**Funding acquisition:** Jin-Woo Park.

**Investigation:** Jiyoun Lee.

**Methodology:** Sunghee Han.

**Project administration:** Jiyoun Lee, Jung-Hee Ryu, Soo Hyun Seo, Jin-Woo Park.

**Supervision:** Jung-Hee Ryu, Sunghee Han, Jin-Woo Park.

**Writing – original draft:** Jiyoun Lee, Jung-Hee Ryu, Soo Hyun Seo, Sunghee Han, Jin-Woo Park.

**Writing – review & editing:** Jiyoun Lee, Jung-Hee Ryu, Sunghee Han, Jin-Woo Park.

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
