## [Decision Letter · Decision Letter 0]

8 Mar 2024

PONE-D-24-02483Virtual Reality vs. Tablet Video for Venipuncture Education in Children: A Randomized Clinical TrialPLOS ONE

Dear Dr. Park,

Thank you for submitting your manuscript to PLOS ONE. After careful consideration, we feel that it has merit but does not fully meet PLOS ONE’s publication criteria as it currently stands. Therefore, we invite you to submit a revised version of the manuscript that addresses the points raised during the review process.

We look forward to receiving your revised manuscript.

Kind regards,

Guglielmo Campus, Ph.D DDS

Academic Editor

PLOS ONE

“This work was supported by the National Research Foundation of Korea (NRF) grant funded by the Korea government (MSIT). (No. 2022R1F1A1072210).”

3. We note that you have a patent relating to material pertinent to this article. Please provide an amended statement of Competing Interests to declare this patent (with details including name and number), along with any other relevant declarations relating to employment, consultancy, patents, products in development or modified products etc. Please confirm that this does not alter your adherence to all PLOS ONE policies on sharing data and materials, as detailed online in our guide for authors http://journals.plos.org/plosone/s/competing-interests by including the following statement: "This does not alter our adherence to  PLOS ONE policies on sharing data and materials.” If there are restrictions on sharing of data and/or materials, please state these. Please note that we cannot proceed with consideration of your article until this information has been declared.

4. In the online submission form you indicate that your data is not available for proprietary reasons and have provided a contact point for accessing this data. Please note that your current contact point is a co-author on this manuscript. According to our Data Policy, the contact point must not be an author on the manuscript and must be an institutional contact, ideally not an individual. Please revise your data statement to a non-author institutional point of contact, such as a data access or ethics committee, and send this to us via return email. Please also include contact information for the third party organization, and please include the full citation of where the data can be found.

5. We note that Figure 2 in your submission contain copyrighted images. All PLOS content is published under the Creative Commons Attribution License (CC BY 4.0), which means that the manuscript, images, and Supporting Information files will be freely available online, and any third party is permitted to access, download, copy, distribute, and use these materials in any way, even commercially, with proper attribution. For more information, see our copyright guidelines: http://journals.plos.org/plosone/s/licenses-and-copyright.

1. You may seek permission from the original copyright holder of Figure 2 to publish the content specifically under the CC BY 4.0 license.

Reviewers' comments:

Reviewer's Responses to Questions

**Comments to the Author**

1. Is the manuscript technically sound, and do the data support the conclusions?

Reviewer #1: Yes

Reviewer #2: Yes

2. Has the statistical analysis been performed appropriately and rigorously? 

Reviewer #1: Yes

Reviewer #2: No

3. Have the authors made all data underlying the findings in their manuscript fully available?

Reviewer #1: Yes

Reviewer #2: Yes

4. Is the manuscript presented in an intelligible fashion and written in standard English?

Reviewer #1: Yes

Reviewer #2: Yes

5. Review Comments to the Author

Reviewer #1: I have had the great opportunity to review your article, and overall, it provides valuable insights into your field. However, I would like to request some clarifications and additional details, particularly in the Methods section.

Firstly, could you elaborate more on the study protocol in the Methods section? While it is mentioned that a single-blinded researcher measured the CHEOP score and outcomes, I am curious about the actual study procedures. For instance, did children experience virtual reality or watch conventional videos in separate room and move to phlebotomy unit afterward? Additionally, what was the time interval between education and venipuncture? Also, one minor thing about your supplement data ‘protocol.docx’. While the total number of children is correctly indicated as 90, there seems to be an inconsistency in the allocation to both groups, where each is designated as 30.

Also, to establish the similarity between both groups, I recommend including p-values in Table 1 to support the claim about their comparable characteristics.

I observed that the satisfaction scores from guardians were uniformly high, with most participants seems like giving a score of 10/10 in both groups. Given the limited variability in satisfaction scores, traditional statistical analyses may not be well-suited for such uniform data. I am interested in understanding the statistical approach used to assess and compare these high satisfaction scores. Moreover, considering the near-universal high ratings, it raises a broader question about the meaningfulness of comparing satisfaction levels between the two groups.

Regarding the excluded patients, the figure 1 flowchart displays only those who did not agree to consent, while you mentioned other exclusion criteria. To adhere to the CONSORT guidelines, could you refine the number of excluded children in the flowchart accordingly?

Moreover, I am interested in understanding the reasoning behind excluding children with a history of venipuncture in the previous year. Could you provide further details on this aspect?

Concerning the CHEOP score, I would appreciate it if you could clarify your rationale for dividing the score into three categories (mild/moderate/severe) and provide the reference for this categorization. Additionally, why did you choose the CHEOP score over other pain scale systems for children? Given that the CHEOP score was initially designed for children aged 1 to 5 years old and those who are postoperative, could you provide more explanation and references on why you specifically chose this scoring system?

Thank you for your time and attention to these queries. Your clarification on these points will contribute to a more comprehensive understanding of your study.

Reviewer #2: What test was used for the sample size calculation? Is that a two-sample t-test? Why not use Mann-Whitney as the primary analysis does?

Can you calculate a Spearman correlation between VAS and the pain scale, overall and by education group?

Is there a significant interaction between VAS and group in the regression?

6. PLOS authors have the option to publish the peer review history of their article (what does this mean?). If published, this will include your full peer review and any attached files.

Reviewer #1: No

Reviewer #2: No

---

## [Author Response · Author response to Decision Letter 0]

15 Apr 2024

Reviewer 1

1) Firstly, could you elaborate more on the study protocol in the Methods section? While it is mentioned that a single-blinded researcher measured the CHEOP score and outcomes, I am curious about the actual study procedures. For instance, did children experience virtual reality or watch conventional videos in separate room and move to phlebotomy unit afterward? Additionally, what was the time interval between education and venipuncture? Also, one minor thing about your supplement data ‘protocol.docx’. While the total number of children is correctly indicated as 90, there seems to be an inconsistency in the allocation to both groups, where each is designated as 30.

Answer) 

Thank you for your kind feedback. We performed the interventions in an independent room separated from the phlebotomy unit, and there was an interval of 5 min or less between education and venipuncture. 

We have provided additional explanations in the Methods section as follows. 

“Another independent researcher received a sealed envelope with the randomization number and performed the intervention in an independent room separated from the phlebotomy unit.” (page: 6) 

“Immediately after the VR or video session, the patients were moved to the phlebotomy unit for venipuncture. The interval from the end of education to the positioning at the phlebotomy desk did not exceed 5 min.” (page: 7) 

We have also added the description to “protocol.docx”.

2) Also, to establish the similarity between both groups, I recommend including p-values in Table 1 to support the claim about their comparable characteristics.

Answer) 

Thank you for your suggestion. P-values have been included as advised.

3) I observed that the satisfaction scores from guardians were uniformly high, with most participants seems like giving a score of 10/10 in both groups. Given the limited variability in satisfaction scores, traditional statistical analyses may not be well-suited for such uniform data. I am interested in understanding the statistical approach used to assess and compare these high satisfaction scores. Moreover, considering the near-universal high ratings, it raises a broader question about the meaningfulness of comparing satisfaction levels between the two groups.

Answer) 

Thank you for the advice. In fact, we drew attention to the fact that the results were contrary to those of a previous study that showed a significant difference in parental satisfaction between the VR group and the control group which received simple verbal education. Compared using different approaches, there was no significant difference in the parental satisfaction between the two groups. Active multimedia education to reduce procedural pain in pediatric patients might increase the satisfaction of the parents, in both groups. We added the description in the discussion section as follows.

“These results are contrary to those of a previous study that showed a significant difference in parental satisfaction between the VR group and the control group which received simple verbal education [15]. Active multimedia education to reduce procedural pain in pediatric patients might increase the satisfaction of the parents, in both groups.” (page: 14, line: 5-9) 

4) Regarding the excluded patients, the figure 1 flowchart displays only those who did not agree to consent, while you mentioned other exclusion criteria. To adhere to the CONSORT guidelines, could you refine the number of excluded children in the flowchart accordingly?

Answer)

Thank you for your kind clarification. Upon review, I have found that the figure missing the excluded patients has been incorrectly registered. I will attach the newly revised Figure 1 flow chart. The "Patient" section in the Methods has also been updated accordingly. Thank you.

5) Moreover, I am interested in understanding the reasoning behind excluding children with a history of venipuncture in the previous year. Could you provide further details on this aspect?

Answer) 

We appreciated the comment. When we planned the research, we hypothesized that if children had prior experience with the same procedure within the last year, it could potentially influence their perception of pain and anxiety associated with the procedure, either positively or negatively. To minimize any bias, children with the history were excluded from the study, as we did in our previous studies regarding VR experiential education.

6) Concerning the CHEOP score, I would appreciate it if you could clarify your rationale for dividing the score into three categories (mild/moderate/severe) and provide the reference for this categorization. Additionally, why did you choose the CHEOP score over other pain scale systems for children? Given that the CHEOP score was initially designed for children aged 1 to 5 years old and those who are postoperative, could you provide more explanation and references on why you specifically chose this scoring system?

Answer)

Thank you for the comment. In our current study, it was revealed that there is a statistically significant difference in CHEOP scores between the two groups. While this statistical difference is important, we also divided the data into thirds to visually demonstrate the distribution of pain among the children. There is no specific rationale for this division.

The CHEOPS (1) was developed for use in the postoperative setting for children one to 12 years of age, but has been applied to other acute pain situations, such as immunizations (2,3), venipuncture (4,5), fracture reduction (6) and laceration repair (7). CHEOPS has included the age range of our study, and is one of the most used behavioral assessment tools to evaluate pain caused by venipuncture in previous research. Accordingly, we decided to use CHEOPS in this study, as we did in our previous study (8).

1. McGrath PJ, Johnson G, Goodman JT, Dunn J, Chapman J. CHEOPS: A behavioral scale for rating postoperative pain in children. In: Fields HL, Dubner R, Cervero F, editors. Advances in Pain Research and Therapy. New York: Raven Press; 1985. pp. 395–402. 

2. Cassidy KL, Reid GJ, McGrath PJ, et al. Watch needle, watch TV: Audiovisual distraction in preschool immunization. Pain Med. 2002;3:108–18. 

3. Cassidy KL, Reid GJ, McGrath PJ, Smith DJ, Brown TL, Finley GA. A randomized double-blind, placebo-controlled trial of the EMLA patch for the reduction of pain associated with intramuscular injections in four to six-year-old children. Acta Paediatr. 2001;90:1329–36.

4. Galinkin JL, Rose JB, Harris K, Watcha MF. Lidocaine iontophoresis versus eutectic mixture of local anesthetics (EMLA) for IV placement in children. Anesth Analg. 2002;94:1484–8.

5. Van-Cleve L, Jonson L, Pothier P. Pain responses of hospitalized infants and children to venipuncture and intravenous cannulation. J Pediatr Nurs. 1996;11:161–7.

6. McCarty EC, Mencio GA, Walker LA, Green NE. Ketamine sedation for the reduction of children’s fractures in the emergency department. J Bone Joint Surg Am. 2000;82:912–8.

7. Klein EJ, Diekema DS, Paris CA, Quan L, Cohen M, Seidel KD. A randomized, clinical trial of oral midazolam plus placebo versus oral midazolam plus oral transmucosal fentanyl for sedation during laceration repair. Pediatrics. 2002;109:894–7.

8. Ryu JH, Han SH, Hwang SM, Lee J, Do SH, Kim JH, et al. Effects of Virtual Reality Education on Procedural Pain and Anxiety During Venipuncture in Children: A Randomized Clinical Trial. Front Med (Lausanne) 2022;9:849541.

Reviewer 2

1) What test was used for the sample size calculation? Is that a two-sample t-test? Why not use Mann-Whitney as the primary analysis does?

Answer) 

Thank you for the comment. We used a two-sample t-test to calculate the sample size. This was because that the number of samples in our study was 45 people per group, so we expected CHEOPS score to show normal distribution. However, the variable did not satisfy normal distribution, so the Mann-Whitney test was used for statistical analysis instead. Of course, even if Mann-Whitney was used to calculate the sample size, there would not have been a sufficient difference in the total number of participants, and the statistical results would not have been different from the current results.

2) Can you calculate a Spearman correlation between VAS and the pain scale, overall and by education group? Is there a significant interaction between VAS and group in the regression?

Answer)

 As you recommended, we performed Spearman correlation analysis between baseline pain expectation (VAS) and the CHEOPS. We found a positive and significant correlation; the Spearman correlation coefficient was 0.477 (P < 0.001). We also performed Spearman correlation analysis between overall and by education group. Except for CHEOPS, no variables showed significant correlation with education group. The Spearman correlation coefficient between the education group and pain expectation (VAS) was -0.012 (P = 0.912).

---

## [Decision Letter · Decision Letter 1]

7 Jul 2024

Virtual Reality vs. Tablet Video for Venipuncture Education in Children: A Randomized Clinical Trial

PONE-D-24-02483R1

Dear Dr. Park,

We’re pleased to inform you that your manuscript has been judged scientifically suitable for publication and will be formally accepted for publication once it meets all outstanding technical requirements.

Kind regards,

Cho Lee Wong, PhD

Academic Editor

PLOS ONE

Additional Editor Comments (optional):

Reviewers' comments:

Reviewer's Responses to Questions

**Comments to the Author**

1. If the authors have adequately addressed your comments raised in a previous round of review and you feel that this manuscript is now acceptable for publication, you may indicate that here to bypass the “Comments to the Author” section, enter your conflict of interest statement in the “Confidential to Editor” section, and submit your "Accept" recommendation.

Reviewer #1: All comments have been addressed

Reviewer #2: All comments have been addressed

2. Is the manuscript technically sound, and do the data support the conclusions?

Reviewer #1: Yes

Reviewer #2: (No Response)

3. Has the statistical analysis been performed appropriately and rigorously? 

Reviewer #1: Yes

Reviewer #2: (No Response)

4. Have the authors made all data underlying the findings in their manuscript fully available?

Reviewer #1: Yes

Reviewer #2: (No Response)

5. Is the manuscript presented in an intelligible fashion and written in standard English?

Reviewer #1: No

Reviewer #2: (No Response)

6. Review Comments to the Author

Reviewer #1: (No Response)

Reviewer #2: All my comments are addressed.

The statistics is acceptable.

7. PLOS authors have the option to publish the peer review history of their article (what does this mean?). If published, this will include your full peer review and any attached files.

Reviewer #1: No

Reviewer #2: No

---

## [Editor Report · Acceptance letter]

17 Jul 2024

PONE-D-24-02483R1 

PLOS ONE

Dear Dr. Park, 

I'm pleased to inform you that your manuscript has been deemed suitable for publication in PLOS ONE. Congratulations! Your manuscript is now being handed over to our production team.

Kind regards, 

on behalf of

Dr. Cho Lee Wong 

Academic Editor

PLOS ONE